# Transport of High-Risk Infectious Substances: Packaging for the Transport of Category A Infectious Specimens in Spain

**DOI:** 10.3390/ijerph192012989

**Published:** 2022-10-11

**Authors:** Jorge H. Sánchez, Susana Gouveia, Claudio Cameselle

**Affiliations:** 1BiotecnIA Group, Department of Chemical Engineering, University of Vigo, 36310 Vigo, Spain; 2HISANTA S.L., 36208 Vigo, Spain

**Keywords:** ADR, UN 2814, UN 2900, infectious substances, Category A

## Abstract

Infectious specimens and materials with pathogens included in Category A of the European Agreement concerning the International Carriage of Dangerous Goods by Road (ADR) must be transported following Packing Instruction P620. A triple packaging system must include leakproof receptacles and impact-resistant packaging to preserve the integrity of the samples and prevent the release of their content in any event during transport. ADR Packing Instruction P620 indicates that the primary receptacle or secondary packaging must withstand, without leakage, an internal pressure not less than 95 kPa at temperatures ranging from −40 °C to +55 °C. This study analyzes various packaging systems available in the Spanish market for the transportation of infectious samples to determine if they comply with the overpressure test, the most difficult to meet according to Packing Instruction P620. Five packaging systems were selected in this study. None of the secondary packaging tested showed adequate characteristics to withstand the pressure leakproof test. In this case, a primary receptacle (containing the sample directly) capable of withstanding an internal pressure of 95 kPa without leakage must be used (for example: test tubes with screw caps). However, manufacturer or distributor specifications are not always clear or readily available in this regard. Health, laboratory, and carrier personnel should be aware of the ADR regulation and packaging characteristics for safe and secure handling and transportation of high-risk Category A infectious materials.

## 1. Introduction

Pathogens are microorganisms that can cause disease in animals or humans. They are usually classified as follows: bacteria, viruses, rickettsia, parasites, and fungi. Pathogens also include other agents such as prions, which became public knowledge in the 1990s due to bovine spongiform encephalopathy, also known as mad cow disease. Infectious substances, for the purpose of transport, are materials or products which are known to contain, or are reasonably expected to contain, biological agents that cause disease in humans or animals (i.e., pathogens) [1]. In this context, the terms “infectious substances”, “infectious materials”, or “infectious products” are considered to be synonymous.

The World Health Organization (WHO) classifies infectious microorganisms according to risk groups based on the degree to which they cause injury through disease, with Risk Group 1 presenting the lowest risk and Risk Group 4 presenting the highest risk [1]. This classification was transposed to Spanish regulations in Royal Decree 664/1997 [2], which classifies biological agents into four groups according to their risk of causing infection or injury through disease. Group 1 includes pathogens with low probability of causing disease in humans or animals. Group 4 includes biological agents that have a high probability of being transmissible from one individual to another due to their ability to cause serious disease in humans or animals. Since there is usually no effective prophylaxis or treatment, they are considered high risk for individuals and the population. Some examples of risk Group 4 pathogens include Ebola virus, Marburg virus, smallpox virus, and others. Handling of samples or substances containing Group 4 biological agents must be done in laboratories with the highest Biosafety Level 4 (BSL-4) [3].

As far as transport is concerned, infectious substances are considered dangerous goods, and specific regulations apply depending on the means of transport chosen. For example, the European Regulation concerning the International Carriage of Dangerous Goods by Rail (RID) applies to rail transport, the International Maritime Dangerous Goods Code (IMDG Code) to sea transport, and the “Technical Instructions for the Safe Transport of Dangerous Goods by Air” of the ICAO (International Civil Aviation Organization) or the “Dangerous Goods Regulations” of the IATA (International Air Transport Association) to air transport.

The transport of dangerous goods by road, including infectious substances, is subject to the provisions of the “European Agreement Concerning the International Carriage of Dangerous Goods by Road (ADR)”. This regulation, which has been regularly updated since its first edition in 1957, contains basic regulation on the minimum conditions for the transport of dangerous substances by road [4,5]. This regulation governs the packaging, transportation, documentation, and other aspects of the transport of dangerous goods by road, including loading, unloading, and storage during transport. An important aspect of this regulation is the definition of the duties and responsibilities of all parties involved in the transport, with the aim of preventing damage to people and property and protecting the environment. The regulation concerns both those directly involved in the transport and the manufacturers of elements and materials related to the transport, packaging, and handling of dangerous goods [6,7].

According to ADR regulation [4,5], infectious substances are classified as “Class 6.2” hazardous goods. If such infectious substances are carried in a form that may cause permanent disability or fatal or life-threatening disease in previously healthy humans or animals upon exposure, they are classified as Category A, and they are assigned the appropriate UN number for transportation as shown in Table 1, depending on whether the specimens contain human pathogens, animal pathogens, or clinical wastes. Category A pathogens are listed in Chapter 2.2.62 of the ADR [4,5]. Those infectious substances, samples, or wastes that do not meet the criteria for Category A are classified as Category B, which includes those substances that may cause disease but do not pose a threat to human or animal life.

Safe packaging systems consisting of three elements are used when transporting Category A infectious substances [8]. The primary receptacle is usually a test tube or cup with a lid containing the infectious materials. The secondary packaging is usually a bag, cylinder, or box containing the primary receptacle. Finally, the tertiary or outer package is a closed box or package for transport (Figure 1). Packaging systems for Category A materials must comply with Packing Instruction P620 [4,5], which includes the following tests: Drop resistance from a height of 9 m, puncture test, water spray test, stacking and, most importantly, the ability of the primary receptacle or secondary packaging to be able to withstand a minimum internal pressure of 95 kPa without leakage at temperatures between −40 °C and +55 °C. To perform these tests, the standard UNE-EN-ISO 16495:2014 or ISO 16495:2022 can be used, which defines the test procedures with which the containers and packaging for the transport of dangerous goods must comply [9,10]. Although this standard is not mandatory, its results are generally accepted by the administration and the entities involved in the transport of infectious substances. The packaging requirements for clinical waste and Category B specimens are less restrictive (Table 1) [4,5].

In detail, the five resistance tests as per Packing Instruction P620 from the ADR [4,5] are as follows:(a)The drop test assesses the resistance to free-fall drops from a height of 9 m onto a non-resilient, horizontal, flat, massive, and rigid surface.(b)The puncture test assesses the resistance to perforation from an impact with a steel rod of 7 kg.(c)If the external package is made of cardboard/fiberboard, it must be subjected to the water spray test simulating rain fall (two inches per hour) applied to the four sides of the package for 1 h to assess the stability and integrity of the packaging system for 1 h.(d)The packaging system must withstand a load equivalent to the total mass of identical packages stacked up to a minimum height of 3 m for 24 h.(e)The primary receptable or the secondary packaging must be capable of withstanding, with no leaking, an internal differential pressure of 95 kPa for 30 min.

The World Health Organization highlighted the importance of good practices in the handling of infection samples [8] and the use of the triple packaging system for their transportation [1]. In general, laboratory personnel are aware of the risk of collection, handling, and analysis of infections samples. They have the initial responsibility for properly classifying, labeling, and packaging infectious specimens [11]. Proper packaging and shipment reduce the risk for all the actors in the transportation chain [12]. However, not all the actors in the transportation have a complete knowledge of the regulations or have received the training to be aware of the risks associated with the handling of infectious materials. For instance, Guilinger et al. [13] identified some limitations in the packaging materials such as: packaging system cost, leaking of samples, and rejection of packages for improper labeling and packaging. Guilinger et al. [13] suggest the use of a deactivation material for infectious substances in case of breakage or leaking. Thus, the transport of infectious substances can be done with very low risk with an acceptable cost.

Táboas et al. [14] compared the ADR requirements with the biosafety practices in a laboratory. As a conclusion, these authors claim that laboratories do not have good control over their own Category A infectious substances during shipment. Such loss of control increases the probability of security incidents, either accidental or intentional (terrorist attack) to an unacceptable level. These authors suggested the development of triple package systems that strictly comply with all the resistance tests from ADR [4,5] and WHO [1,8] recommendations, with the objective of maintaining the integrity of the samples under any conditions during transportation.

The packaging materials available in the market for the transportation of infectious substances comply with the specifications of the ADR, as reported in the technical information from the manufactures. However, the breakage of containers and leaking of infectious specimens during transportation due to fails in the primary receptacle or the secondary packaging have been reported [15]. Such biosafety fails increase risk and compromise human and animal health.

Various research studies [6,7,16,17] are available in the literature about the characteristics of the packaging for Category A and B infectious substances, as described in the international regulations [1,5]. Some publications describe the risk associated with failures in packaging materials during transportation [15], but there are no available studies about the resistance of the triple packaging systems in relation to the risks for all the actors in the handling and transportation of infectious materials.

The aim of this work is to evaluate the characteristics of the packaging systems for the transportation of infectious substances available on the Spanish market. The final objective is to assess whether these containers comply with the packaging regulations, paying special attention to the overpressure leakproof test at 95 kPa, since it is the most difficult resistance test with which to comply.

## 2. Materials and Methods

### 2.1. Packaging

Five types of secondary and tertiary packaging available on the Spanish market for shipping Category A infectious substances were selected for this study. All of the selected packaging types are commercially marked as suitable for the carriage of dangerous goods in a primary receptacle. The manufacturers or distributors of the selected packaging systems in this study do not supply a specific primary receptacle or provide information about what primary receptacle must be used, so it is assumed that it is the user’s responsibility to select the primary receptacle that best suits the characteristics of the infectious specimen. As per the ADR, the packaging systems for Category A infectious substances must comply with the following resistance tests: free-fall drop, puncture test, water spray test, stacking up to 3 m, and overpressure test at 95 kPa. Among these tests, the most difficult to comply with is the overpressure at 95 kPa. Moreover, the overpressure test is related to the breakage of containers and the leaking of infectious substances during transportation as reported in the literature [15]. In this study, the pressure leakproof test was performed on each of the selected secondary containers shown in Figure 2.

(a)Packaging 1

This consists of a tertiary or outer package, which is a compact cardboard cylinder with a layer of expanded polystyrene on the inner wall (Figure 2 (1)). Its outer diameter is 100 mm with a height of 172 mm. The secondary container (located inside the previous one) consists of a compact cardboard cylinder with a deep-drawn metal bottom with an outer diameter of 67 mm and a total height of 150 mm. Both secondary and outer packaging have a movable lid closure on the top of the cylinder, which is made of rigid plastic, has no seal, and is connected to the cylinder body under pressure. Secondary packaging has a compact cardboard core inside the cylinder. This mandrel houses the primary receptacle that contains the infectious specimen.

In the data sheet, the packaging is described as suitable for transporting all types of infectious biological specimens (both Category A and Category B) that comply with Packing Instructions P620 and P650 from the ADR.
(b)Packaging 2

This consists of a tertiary or outer package, which is a pressed corrugated box with an external closure system (Figure 2 (2)). The external and internal dimensions are 170 × 170 × 220 mm and 65 × 165 × 205 mm, respectively. Inside this package is the secondary container: a polypropylene drum with a height of 185 mm and a diameter of 140 mm with a capacity of 1 L. It has a screw cap with an O-ring. A sheet of absorbent material is included.

According to the manufacturer, the secondary packaging is intended for transport of diagnostic samples and infectious substances (CLASS 6.2 in the ADR). It complies with ADR Packing Instructions P620 and P650 and is approved in accordance with applicable national and international regulations (ICAO/IATA, ADR, RID, IMDG).
(c)Packaging 3

This is a manufactured outer package made of a double-layered, corrugated cardboard. The external dimensions are 165 × 150 mm, and the total height is 205 mm (Figure 2 (3)). It has a double-sided tape closure. The secondary container is held inside. It is a plastic drum with a nominal volume of 1.8 L. Its outer diameter is 158 mm, and its total height is 186 mm. It is equipped with a screw cap made of polypropylene. A sheet of absorbent material is included.

The manufacturer of Packaging 3 states that the container is suitable for transporting infectious substances with UN numbers 2814 and 2900 under the conditions of a maximum gross mass of 0.7 kg and a maximum stacking height of 3 m.
(d)Packaging 4

Packaging 4 (Figure 2 (4)) is a polypropylene box with external dimensions (length, width and height) of 270 × 125 × 180 mm. It has a body with a sealing ring and a lid closed with 2 tabs provided with eaves on the shorter sides of the top. Inside this box, there is a rigid foam structure that can hold up to 40 primary receptacles. Under this structure, a film of absorbent material covers the bottom of the package. On the product data sheet, the manufacturer specifically recommends what type of primary packaging to use. It also indicates that the container is liquid-tight once it contains foam as the absorbent material.
(e)Packaging 5

Packaging 5 (Figure 2 (5)) consists of an isothermal external tertiary container made of ABS plastic with dimensions (height, length, and width) of 414 × 504 × 387 mm. The container consists of a body with a sealing ring and a lid closed by 4 bracket-like tabs on each side of the body, which snap into the lid by pressure. The secondary containers are held inside. They are plastic boxes with dimensions of (height, length and width) 140 × 266 × 142 mm. These boxes consist of a body and a lid with a clip closure on the shorter sides of the lid. They include a foam structure to hold the primary receptacles with the infectious specimen. The tertiary outer container can hold up to 6 units of the secondary containers. Stiff cold accumulators can be placed in the external container lid to preserve the specimens at low temperature.

### 2.2. Pressure Leakproof Test Methodology

In this study, five packaging systems available on the market have been selected for the transport of high-risk infectious substances (Category A), with the aim of determining whether they comply with Packing Instruction P620 from the ADR [4,5]. Specifically, the study has focused on the leakproof test, norm ISO 16495:2022, with an internal pressure of +95 kPa, as it is the most difficult to comply with. All the tests have been carried out at room temperature and always in triplicate to ensure the validity and reproducibility of the results. It has been decided to test the secondary packaging since the selected packaging systems do not include a primary receptacle.

The leakproof tests have been carried out by injecting pressurized air into the secondary container submerged in a tank full of water. The containers were filled with lead weights to keep them submerged during the tests. In this way, whether or not the packages were leakproof could be determined visually by the formation of bubbles in the event of leaks. The air was injected using a compressor (SENCO, model: PC1010N) with a maximum output of 860 kPa, through a threaded connector that was installed on the lid of each container to be tested (Figure 3). A T-type union was installed on this connector to simultaneously connect the air supply from the compressor and a pressure gauge to measure the pressure at the container inlet. The compressor had a pressure reducer at the air outlet that allowed the adjustment of pressure supplied to the container. All connections were checked to ensure that there were no air leaks and that the gauge reading was stable and reliable. The maximum pressure supported by each of the containers before the detection of air leaks was recorded, as well as the time that it remained at that maximum registered pressure.

Air humidity and temperature during the tests were determined using a TES-1360 multiparameter probe (TES Electrical Electronic Corp., Taiwan). The environmental conditions in the laboratory during the tests were between 15.3 and 16.6 °C and between 40.0 and 44.1% relative humidity.

## 3. Results

### Pressure Leakproof Test

(a)Packaging 1

When pressurized air was supplied to the closed container immersed in a water tank, the formation of numerous air bubbles at the junction between the lid and the container was clearly evident after 4 s of testing. The lid became detached when an overpressure of 12 kPa was reached, which is still far from the 95 kPa pressure specified in the standard. This test was repeated out of the water tank to avoid any negative effect of water on the carboard of the secondary packaging. We obtained the same result. The lid popped off at 12 kPa. From these tests, it can be concluded that the secondary packaging of Packaging 1 without primary receptacle does not meet the air tightness test.
(b)Packaging 2

The air tightness test was clearly unsatisfactory, as leaks occurred immediately after pneumatic pressure was supplied into the interior of the container when the pressure gauge showed zero. The packaging remained intact until the required pressure of 95 kPa, but significant bubbling was observed in the area where the lid joined the container throughout the duration of the test.
(c)Packaging 3

The secondary container of Packaging 3 was pressure tested at room temperature at 95 kPa. The first container tested had the lid closed to the mark indicated on the container itself for the application of the security seals. Under these conditions, air leakage was detected 2 s after the start of the test, without exceeding an internal pressure of 20 kPa. When testing the second container, the lid was tightened as much as possible to see if such a closure would improve the tightness results. Air loss and abundant bubbling were noted in the water bath 8 s after the test began. The internal pressure did not exceed 52 kPa. The third test was performed under the same conditions as the previous one, with the lid closed to the maximum. In this case, the results were slightly lower, an air leak was detected after 4 s, with the internal pressure not exceeding the value of 40 kPa.

Considering the variability of the results, another 3 units of the same model were tested (a total of six replicates of package 3). In these three additional tests, it was found that air leaked after 2 s without reaching 10 kPa of pressure for the first container. The second one leaked when the internal pressure reached 32 kPa. Finally, the third container showed no leak of air for 30 s at 95 kPa.

Although one of the tested containers was able to maintain tightness for a short period of time (30 s), the overall result for Packaging 3 is that it does not meet the requirements of the pressure test at +95 kPa.
(d)Packaging 4

In the tests of Packaging 4, leaks were observed in the three tested elements immediately after the introduction of pneumatic pressure inside the container. The visual analysis of the type of closure of the box already showed a lack of tight closure, and the tests performed showed the ineffectiveness of the sealing ring included in the package (Figure 2 (4)).
(e)Packaging 5

The secondary container of Packaging 5 was subjected to 95 kPa pressure test at room temperature. This test was also unsatisfactory, as leaks occurred in all samples immediately after the introduction of pneumatic pressure inside the container (Figure 4).

## 4. Discussion

According to the ADR, packaging for road transport of Category A infectious materials must comply with Packing Instruction P620. This applies to UN 2814 (infectious substances for humans) and UN 2900 (infectious substances for animals only) numbers. This packing instruction is the same as that for other means of transportation (RID, IMDG, ICAO and IATA). As shown in Figure 1, the triple packaging system must consist of at least three elements: a leakproof primary receptacle that holds the sample, a closed secondary container, and a rigid outer or tertiary packaging. The primary receptacle or the secondary packaging shall be capable of withstanding, without leakage, an internal pressure of 95 kPa. This primary receptacle or secondary packaging shall also be capable of withstanding temperatures in the range of −40 °C to +55 °C. In addition, the secondary packaging must contain an absorbent material to collect the liquid fluids in case of breakage or leakage of the primary receptacle. Packing Instruction P620 [4,5] specifies that packaging must meet the five resistance tests described in the materials and methods section [9,10]:

Compliance with Packing Instruction P620 largely depends on packaging design and the material used to make the primary receptacle and the secondary and tertiary packages. A visual preliminary analysis of the packaging available on the market allow us to conclude that the leak test is the most difficult to meet, while the other tests are easier to meet with the selection of appropriate materials and a packaging design that allows for stable and secure assembly.

The information available in the bibliography shows that, despite the strict regulations for the transport of hazardous specimens and infectious materials, leakage of specimens or rupture of containers is sometimes observed, compromising safety in the transport. For example, Graf-Gruber [15] highlighted the use of primary packaging (bags or test tubes) that cannot withstand the pressure differences that can easily occur during air transportation. This leads to leakage of samples that are not retained in the secondary container, either because there is not enough adsorbent material, the packaging is not tightly closed, or it cannot withstand the +95 kPa pressure (Figure 5). Similarly, bursting of containers due to pressure differentials was also observed, indicating the need to test the materials and the design of containers to ensure their integrity under potential pressure changes that may occur during transport.

Poor practices in the transport of specimens and infectious waste were also occasionally reported [18,19,20,21,22] with the use of non-approved packaging, the combination of specimens of different types in the same packaging, the lack of use of the secondary package, or even the improper transport of specimens increasing the risk of breakage (Figure 6). For example, the use of unsafe vehicles (e.g., private vehicles or public transport) for infectious specimens is inappropriate as is carrying specimens in toolboxes, camping coolers, or sports bags as tertiary packaging. Another example of poor practice is the transport of specimens in approved containers but outside the vehicle, with the risk of breaking the container and dispersing its content [20].

Packaging systems on the market for the transport of infectious specimens generally have sufficient characteristics for their safe and secure handling. However, the +95 kPa overpressure test showed that none of the five packages studied had the necessary characteristics to pass this assay (Figure 4, Table 2). Furthermore, it must be taken into account that the leakproof test was carried out at room temperature, when the norm [9,10] requires to pass this test in a temperature range from −40 °C to +55 °C. In other words, none of the five packagings passed the test even in less demanding environmental conditions than those the standard requires.

The international regulations [4,5] require that the internal overpressure test must be passed by the primary receptacle or the secondary container. In case the secondary packaging fails the test, the user must be warned that it is mandatory to use primary receptacles that can withstand the internal pressure of +95 kPa. The primary receptacles shown in Figure 7 are commonly used for transporting infectious specimens. However, not all test tubes or bags pass the internal pressure test at +95 kPa. For example, test tubes with stoppers, such as those shown in Figure 7a, do not pass the pressure test. Simple zip-lock plastic bags often leak at the zip-lock or side edge due to poor sealing. Under these conditions, the user must have sufficient knowledge of the regulations and the characteristics of the primary receptacles to select those receptacles that meet the packaging regulations for the infectious specimens, e.g., test tube with screw cap or plastic bags with flap and double zip-lock (Figure 7b). The manufactures of these primary receptacles have reported their hermetic tightness under +95 kPa overpressure conditions.

## 5. Conclusions

The results in this study have shown that five secondary containers/packagings available in the Spanish market for the transport of Category A infectious specimens did not pass the leak test at +95 kPa at room temperature. In all cases, the injection of compressed air into the secondary container/packaging resulted in air leaks at pressure far below +95 kPa. In some cases, the air leaking occurred at zero-gauge pressure. The lack of tightness in the packaging was mainly due to the design of the container and lid, as well as the properties of the manufacturing materials that also contributed to the lack of tight closing. If the secondary packaging fails the pressure test, the primary receptacle must pass the pressure test, as is described in the ADR. Therefore, the user must know the transport regulations applicable to infectious specimens, and they must be aware of the properties of the packaging used to select those primary receptacles (e.g., test tubes or plastic bags) that meet the overpressure test at 95 kPa for the safe and secure transport of Category A infectious specimens. The five packaging systems tested in this study were selected from the Spanish market, so the conclusions of this study apply to Spain. However, considering that most of the packaging systems tested are also available in the international market, the conclusions of this study may be extended to the transportation of infectious substances at the international level.

## Figures and Tables

**Figure 1 ijerph-19-12989-f001:**
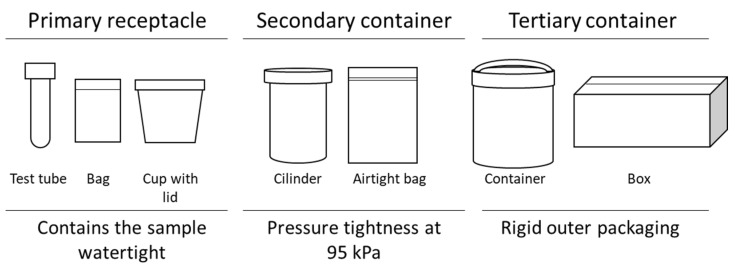
Category A infectious specimen packaging as per P620 from the ADR [4,5].

**Figure 2 ijerph-19-12989-f002:**
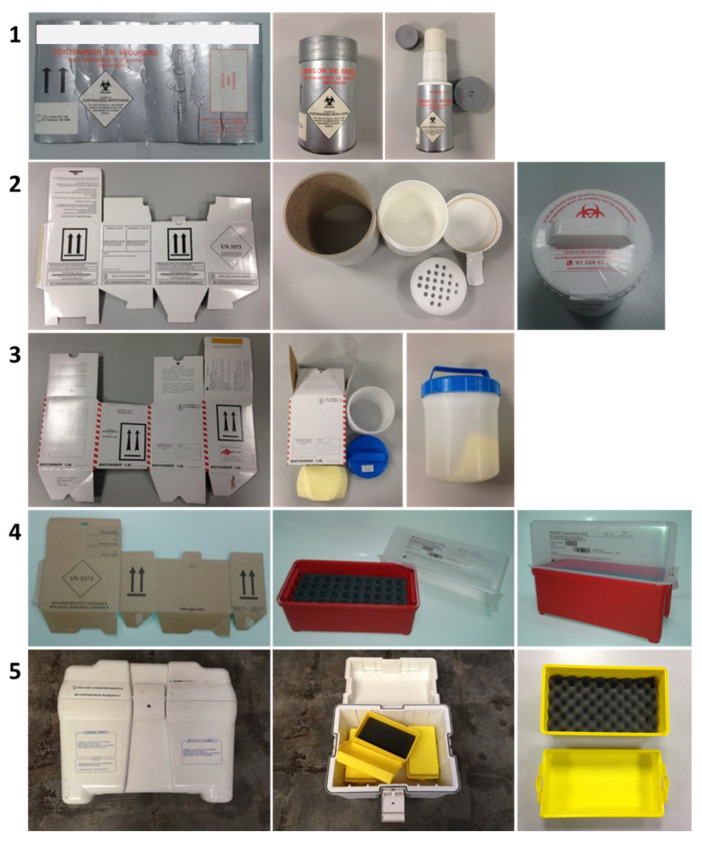
Commercial packaging for the transport of Category A infectious substances.

**Figure 3 ijerph-19-12989-f003:**
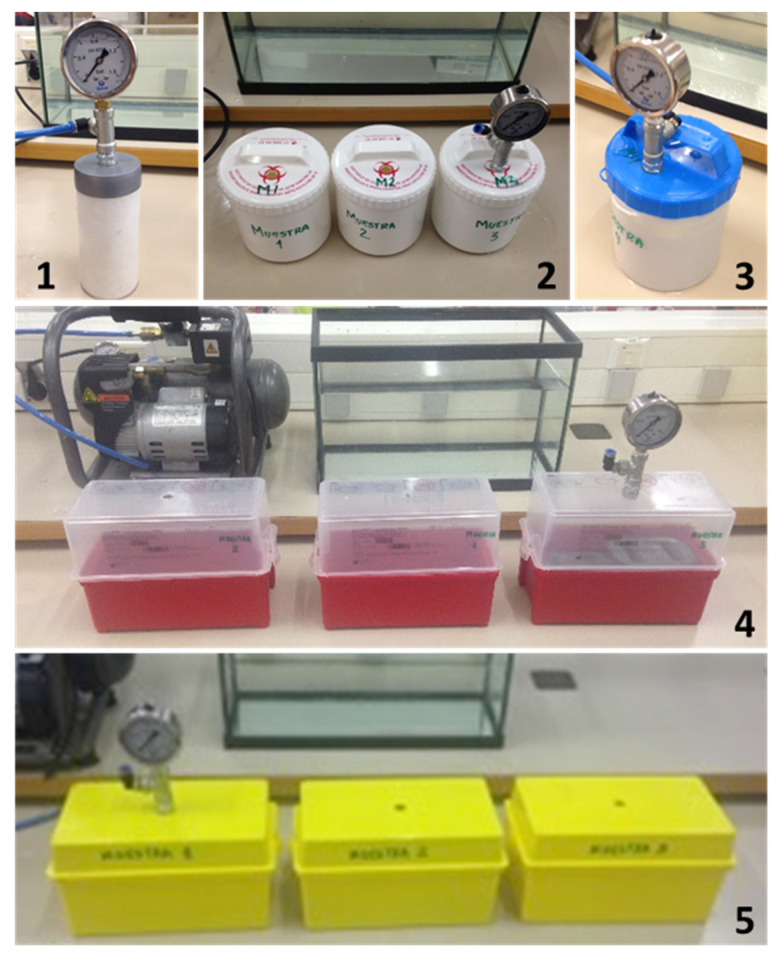
Packaging system with air pressure connector and pressure gauge before the pressure leakproof tests.

**Figure 4 ijerph-19-12989-f004:**
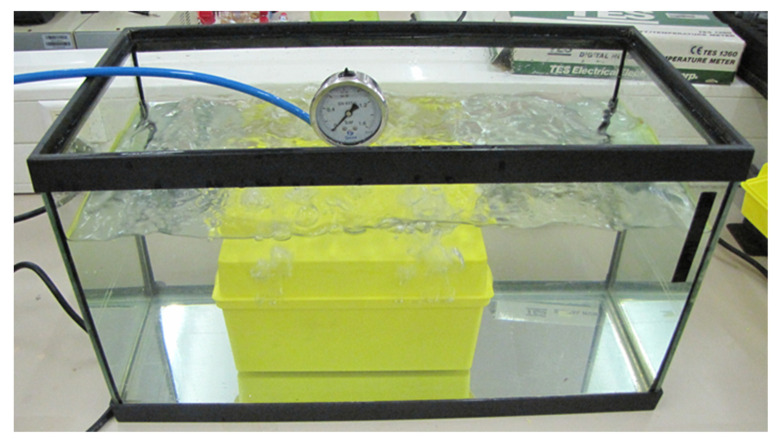
Testing the pressure tightness of a secondary container (package 5) immersed in water.

**Figure 5 ijerph-19-12989-f005:**
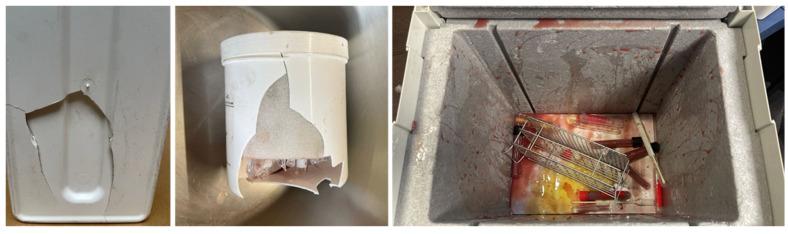
Leaks and breakage of containers in the transport of infectious samples.

**Figure 6 ijerph-19-12989-f006:**
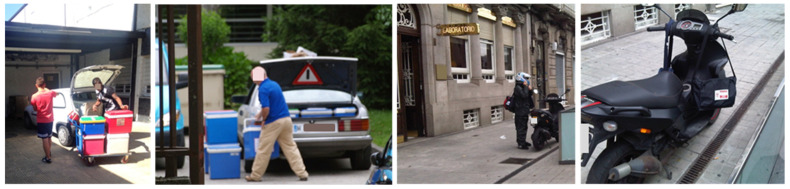
Poor practices in the transport of infectious samples.

**Figure 7 ijerph-19-12989-f007:**
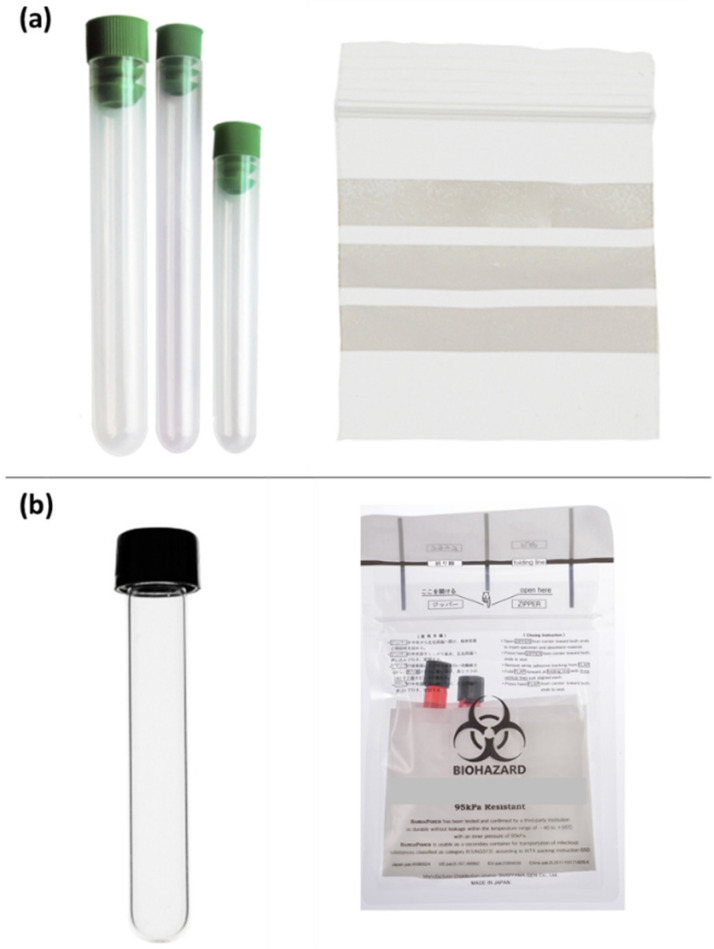
Primary receptacles for infectious samples (**a**) not meeting the test and (**b**) meeting the test of overpressure of +95 kPa.

**Table 1 ijerph-19-12989-t001:** Infectious substances classification and packing instructions according to the ADR [4,5].

Infectious Substances		UN Number	Packing Instruction
Category A	Substances containing human pathogens	UN2814	P620
	Substances containing animal pathogens	UN2900	P620
	Clinical waste	UN3549	P622
Category B	Specimens or biological substances	UN3373	P650
	Waste	UN3291	P621

**Table 2 ijerph-19-12989-t002:** Results from the pressure leakproof tests.

		Resistance with No Leaking		
Packaging	Number of Tested Items	Pressure (kPa)	Time (s)	Comments
1	3 in water tank 3 out the tank	12	0	Lid popped off after 4 s
2	3	0	0	Pressure reached 95 kPa but leaking was observed from the beginning
3	6	10–52	2–10	1 item reached 95 kPa for 30 min with no leaking
4	3	0	0	Lid with no tight closing
5	3	0	0	Bubbling from the beginning

## Data Availability

All the data in presented in this article.

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
