# Peer review of "Transport of High-Risk Infectious Substances: Packaging for the Transport of Category A Infectious Specimens in Spain"

_ijerph, 2022, doi:10.3390/ijerph192012989_

Round 1

Reviewer 1 Report

the paper is well written and fulfilled the research framework. the introduction is very informative and gives enough information about the potential risks of highly infectious substances. the materials and methods are well described and give enough information for repeating experiments. the results are well summarized and easy to understand. however, I suggest to the authors compare all the packages. with graphics or pictures of those results, some values could bring benefits and uniformity of conclusions.

Reviewer 3 Report

This manuscript is about analyzing various packaging for infectious samples to determine if they comply with the overpressure test, the most difficult to meet in the P620 instruction. The manuscript has been well written and organized, but within this study, only a simple pressure test was performed and other useful tests and information regarding the capability of the tested packages for transporting high-risk infectious substances have not been mentioned.

Round 2

Reviewer 2 Report

I believe the manuscript has been significantly improved. E.g.,

- new text portions, directly relevant to the MS ideas and results, have appeared in all the sections of MS, from its Abstract to Conclusions and References;

- new Table 2 within the "Results" now arranges the results from the pressure leakproof tests in more relevant fashion;

- several tasks and problems formulated by the authors in a revised version of MS have improved the understanding of novelty, actuality and advantages given by the MS;

- numerous corrections have been made to the version of MS in accordance to recommendations.

Author Response

RESPONSE: The authors appreciate the positive comments from the reviewer 2. We have revised the text of our manuscript to correct minor errors in English usage.

Reviewer 3 Report

Dear authors,

The manuscript has been revised substantially. However, there are only a few more remarks that should be considered:

Table 1. title: Use lower case letters at the beginning of “Substances” and “Classification”.  

Discussion (lines 349-367): Some information (like the list of resistance tests) is presented here which is mentioned in previous parts (i.e., in the introduction section). Please avoid duplication, and revise.

Author Response

Reviewer 3: Comments and Suggestions for Authors

Dear authors,

The manuscript has been revised substantially. However, there are only a few more remarks that should be considered:

Table 1. title: Use lower case letters at the beginning of “Substances” and “Classification”.  

RESPONSE: We have revised the typos in the title as suggested for reviewer 3.

Discussion (lines 349-367): Some information (like the list of resistance tests) is presented here which is mentioned in previous parts (i.e., in the introduction section). Please avoid duplication, and revise.

RESPONSE: As suggested from reviewer 3, we have removed the duplicate information about the resistance test of the packaging material. We have revised the whole manuscript to remove unnecessary duplication of information.